# Psychosocial experiences modulate asthma-associated genes through gene-environment interactions

**Justyna A Resztak[1], Allison K Farrell[2], Henriette Mair-Meijers[1], Adnan Alazizi[1], Xiaoquan Wen[3], Derek E Wildman[4], Samuele Zilioli[5,6], Richard B Slatcher[5], Roger Pique-Regi[1,7]\*, Francesca Luca[1,7]\***

[1]Center for Molecular Medicine and Genetics, Wayne State University, Detroit, United States; [2]Department of Psychology, Miami University, Oxford, United States; [3]Department of Biostatistics, University of Michigan, Ann Arbor, United States; [4]College of Public Health, University of South Florida, Tampa, United States; [5]Department of Psychology, Wayne State University, Detroit, United States; [6]Department of Family Medicine and Public Health Sciences, Wayne State University, Detroit, United States; [7]Department of Obstetrics and Gynecology, Wayne State University, Detroit, United States

**\*For correspondence:**
rpique@wayne.edu (RP-R);
fluca@wayne.edu (FL)

**Competing interests:** The authors declare that no competing interests exist.

**Abstract** Social interactions and the overall psychosocial environment have a demonstrated impact on health, particularly for people living in disadvantaged urban areas. Here, we investigated the effect of psychosocial experiences on gene expression in peripheral blood immune cells of children with asthma in Metro Detroit. Using RNA-sequencing and a new machine learning approach, we identified transcriptional signatures of 19 variables including psychosocial factors, blood cell composition, and asthma symptoms. Importantly, we found 169 genes associated with asthma or allergic disease that are regulated by psychosocial factors and 344 significant gene-environment interactions for gene expression levels. These results demonstrate that immune gene expression mediates the link between negative psychosocial experiences and asthma risk.

## Introduction

Psychosocial experiences have long been recognized to affect human health (*Miller et al., 2009*). Intrapersonal processes (e.g., emotionality [*Pressman et al., 2019*; *Smith et al., 2004*], interpersonal social relationships [*Repetti et al., 2002*; *Robles et al., 2014*]) and broader structural environments (e.g., neighborhood quality and socioeconomic status [SES]; *Gallo and Matthews, 2003*) are all associated with the morbidity and severity of diseases such as asthma (*Harrison, 1998*), cancer (*Meyer and Mark, 1995*), cardiovascular disease (*Everson-Rose and Lewis, 2005*), as well as mortality rates (*Holt-Lunstad et al., 2010*; *Chida and Steptoe, 2008*). Asthma is a chronic inflammatory disease of the respiratory tract that disproportionately affects children (*Moorman et al., 2012*). It is one of the costliest pediatric health conditions (*Weiss et al., 2000*) and a leading cause of school absenteeism (*Akinbami and Centers for Disease Control and Prevention National Center for Health Statistics, 2006*). Financially struggling cities, such as Detroit, are at an especially high risk for asthma morbidity and mortality (*Sullivan et al., 2002*). While environmental and genetic factors lead to the development of asthma and affect the health of children with asthma (*von Mutius, 2000*; *von Mutius, 2009*; *Umetsu et al., 2002*), psychosocial stress is a critical factor contributing to asthma severity (*Wright et al., 1998*; *Wright et al., 2005*; *Shankardass et al., 2009*; *Chen and*

*Miller, 2007*; *Sandberg et al., 2000*). Understanding the biological pathways underlying these associations is crucial to strengthen the causal claims linking psychosocial experiences and health.

The growing field of social genomics investigates how various dimensions of a person's social and psychological environment influence gene expression (*Cole, 2014*; *Slavich and Cole, 2013*; *Cole, 2009*; *Toyokawa et al., 2012*; *Galea et al., 2011*). There is ample evidence for links between gene expression in blood and three major categories of psychosocial experiences: SES (*Chen et al., 2009*), social relationships (*Robles et al., 2018*; *Stanton et al., 2017*; *Powell et al., 2013*), and emotionality (*Farrell et al., 2018*; *Segman et al., 2010*). Beyond single-gene analyses, previous studies in this area (*Slavich and Cole, 2013*; *Cole, 2014*; *Cole, 2009*) identified a pattern of differentially expressed genes referred to as the *conserved transcriptional response to adversity* (CTRA). The CTRA is characterized by increased expression of genes involved in inflammation and decreased expression of genes involved in type I interferon antiviral responses and IgG1 antibody synthesis (*Fredrickson et al., 2013*). However, these studies investigated a limited set of psychosocial experiences and did not resolve whether these pathways are causally linked to health outcomes or rather a consequence of disease status.

Several approaches have been developed for investigating the role of gene expression in complex trait variation (*Nica et al., 2010*; *Marigorta et al., 2017*; *Võsa et al., 2018*; *Nica and Dermitzakis, 2008*). Recently, transcriptome-wide association studies [TWAS] and other Mendelian randomization (MR) approaches have been used to integrate genetic effects on gene expression and on complex traits to establish causal links between a gene and a phenotype (*Gusev et al., 2016*). MR approaches have been developed in epidemiology to examine the causal effect of a modifiable exposure on disease without conducting a randomized trial. MR designs use the genotype association with the two variables of interest to control for reverse causation and confounding. Here, we use this type of approach to connect genes with complex traits. Traditionally genes are annotated to association signals in genome-wide association studies [GWAS] based on physical proximity. TWAS test for an association between gene expression and complex traits, where gene expression is predicted based on genotypes in the GWAS study and independent expression quantitative trait locus (eQTL) data. Notably this goes beyond physical proximity of GWAS signals to genes and establishes a putative mechanism linking genetic variants to complex traits through genetic regulation of gene expression. Very few studies of genetic regulation of gene expression (expression quantitative trait loci, eQTL mapping) in humans have included comprehensive information on psychosocial exposures, and no study to date has been able to determine the likelihood of a causal relationship between psychosocial experiences, gene expression, and asthma. This study aims at filling this gap by combining genetic and well-characterized psychosocial data from a cohort of children with asthma living in Metro Detroit (*Figure 1a*).

The Asthma in the Lives of Families Today (ALOFT) project was established in 2009 to identify the behavioral and biological pathways through which family social environments impact youth with asthma. This study started during the years leading up to Detroit filing for bankruptcy in 2013 and is still ongoing. Detroit started a marked economic recovery in 2016; yet not all population groups and geographic areas have experienced it simultaneously or to the same extent. To analyze the relationship between psychosocial experiences, asthma, and transcriptional regulation, we investigated genome-wide gene expression (RNA-seq) for 251 youth participating in the ALOFT study. For 119 participants, we also collected 53 psychosocial and biological variables (*Supplementary file 1a, b*, *Figure 1—figure supplement 1*). Measures of psychosocial experiences were grouped into three subcategories, indicating SES, social relationships, and emotionality. Psychosocial experiences were captured through subjective and objective measures (e.g., negative affect assessed from daily diaries and recorded audio, respectively), as well as global and daily measures.

## Results

### Psychosocial factors and asthma alter the transcriptome

To denoise and impute psychosocial effects on gene expression for the entire cohort of 251 participants, we developed a new machine learning approach based on generalized linear models with elastic net regularization (GLMnet; *Friedman et al., 2010*) and cross-validation. Using this approach, we derived transcriptional signatures that represent the portion of the transcriptome that correlates

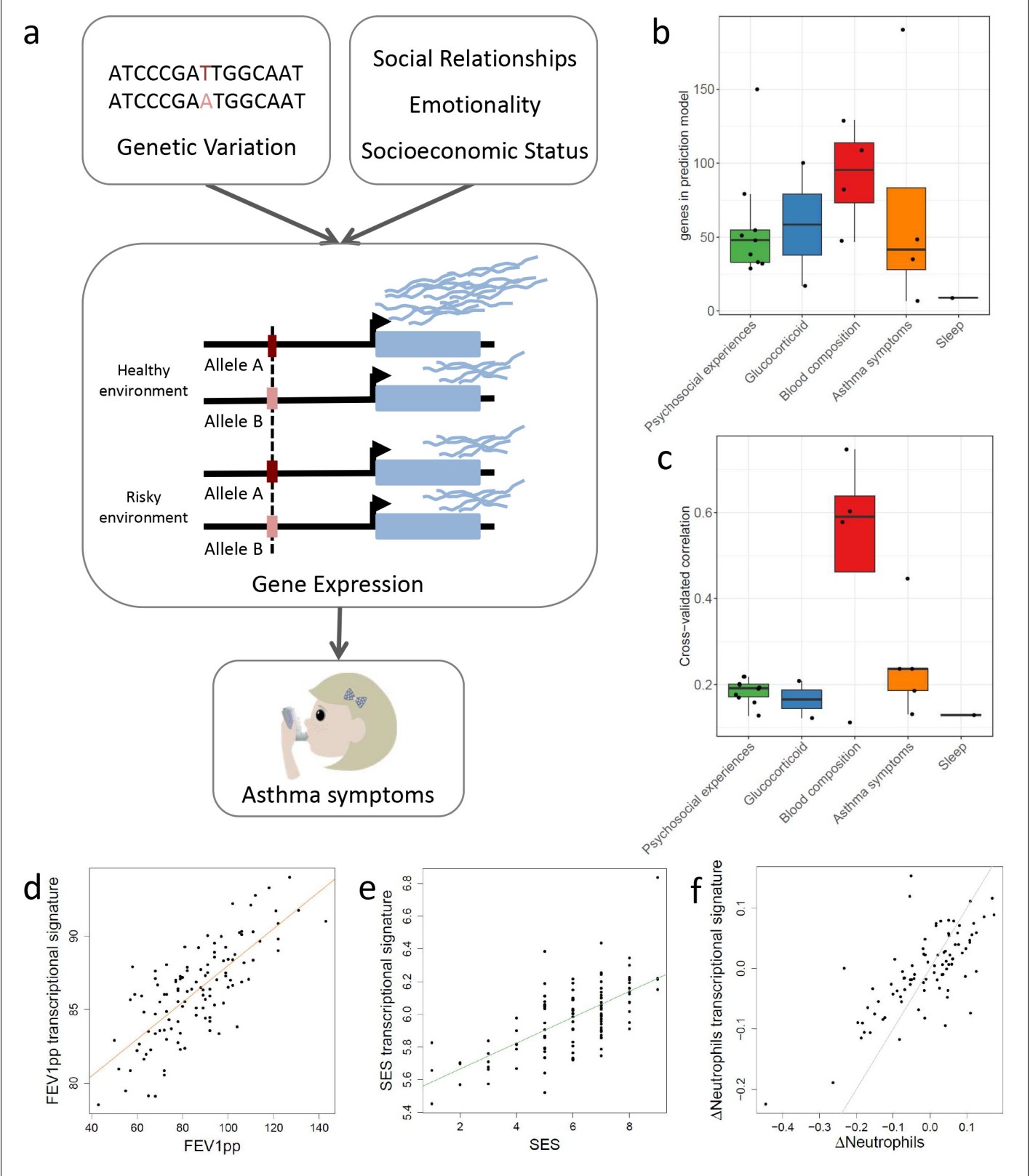

**Figure 1.** Transcriptional signatures of psychosocial experiences and asthma symptoms. (a) Central hypothesis. (b) Number of genes in elastic net regression models that explain at least 1% of variance. Colors represent different categories of variables. (c) Pearson's correlations between cross-validated transcriptional signatures and measured variables for elastic net regression models that explain at least 1% of variance. (d) Forced expiratory

*Figure 1 continued on next page*

*Figure 1 continued*

volume in one second [FEV1] percent predicted transcriptional signature model fit (Pearson's rho = 0.76, p<0.001). (**e**) MacArthur socioeconomic status transcriptional signature model fit (Pearson's rho = 0.67, p<0.001). (**f**) Longitudinal change in observed neutrophils (x axis) and longitudinal change in transcriptional signature of neutrophils (y axis) (Pearson's rho = 0.72, p<0.001, gray = identity line).

The online version of this article includes the following source data and figure supplement(s) for figure 1:

**Source data 1.** GLMnet model weights for the transcriptional signatures.
**Figure supplement 1.** Clustered heatmap of (Pearson) correlations between all variables used in the study.
**Figure supplement 2.** Scatterplot of Pearson's correlation coefficients between each pair of observed variables (x axis) and metagenes (y axis) for the 19 variables with transcriptional signatures explaining >1% of observed variance.
**Figure supplement 3.** Gene set enrichment analysis results on genes differentially expressed for psychosocial experiences.
**Figure supplement 4.** Result of identity-by-descent analysis (IBD; maximum likelihood estimation [MLE]) on DNA-derived genotypes for all 251 participants.
**Figure supplement 5.** Self-reported ethnicity (x axis) vs. percent global African ancestry (y axis) in 119 participants for whom declared ethnicity is available.
**Figure supplement 6.** Proportion of reads mapping to the Y chromosome over all mapped reads separately for self-reported females and males.
**Figure supplement 7.** Sources of variation in gene expression data.
**Figure supplement 8.** Comparison of variance explained by conserved transcriptional response to adversity (CTRA)-based (x axis) and unbiased (y axis) elastic net prediction models; color indicates type of variable (red = identity line).

with each psychosocial factor. Analogous methods have been adopted to define transcriptional signatures of T-cell exhaustion in aging (*Alpert et al., 2019*) and survival in cancer (*Asgharzadeh et al., 2006*), but have not been previously used for psychosocial factors. We identified significant transcriptional signatures for 31 out of 53 variables (*Figure 1b–e*, *Figure 1—source data 1*, *Supplementary file 1c*). We used an independent longitudinal dataset to validate the transcriptional signatures. We considered the changes in the observed variable between two time points ($\geq$1 year) and compared it to the longitudinal changes in the transcriptional signature. Note that the transcriptional signature is imputed for the second time point from gene expression samples that are not included in the training set. We found significant correlations in the observed and imputed changes for the majority of variables (Spearman's correlation p-value<0.05; e.g., *Figure 1f*, *Supplementary file 1d*).

Transcriptional signatures of the SES measures showed a strong overlap with each other (*Figure 2*), suggesting that they may have very similar molecular effects or measure the same factors. However, we also saw correlations across all three variable categories. For example, subjective SES was significantly correlated with objective maternal responsiveness, family conflict, and self-reported self-disclosure, which is the extent to which the youths talk about their thoughts and feelings (r = −0.3, p=1.1 * 10$^{-6}$, r = 0.29, p=4.26 * 10$^{-6}$, r = 0.66, p=3 * 10$^{-32}$, respectively). Overall, correlations between transcriptional signatures reflect correlations between measured variables (*Figure 1—figure supplement 2*), yet they are stronger between the transcriptional signatures, highlighting the denoising effect. Measured psychosocial factors were also associated with interindividual variation in gene expression for several genes. For example, perceived responsiveness and self-disclosure were associated with changes in gene expression for 143 and 3279 genes, respectively (*Supplementary file 1e, f*). Genes positively associated with perceived responsiveness were enriched for biological processes relative to the response to IL18, while genes positively associated with self-disclosure were enriched for the interferon 1 pathway (*Figure 1—figure supplement 3*).

When we correlated transcriptional signatures of asthma severity with those for psychosocial variables, we observed overlap with SES and social relationships, but not emotionality. In particular, we found significant positive correlations between the transcriptional signatures of lung function (percent-predicted FEV1) and psychosocial measures of self-disclosure (r = 0.42, p=4.9 * 10$^{-12}$) and subjective SES (r = 0.36, p=3.3 * 10$^{-9}$). Unexpectedly, objective maternal responsiveness was negatively correlated with lung function (r = −0.19, p=0.002), and percent unoccupied properties in the neighborhood (r = 0.41, p=7.8 * 10$^{-12}$) was positively correlated with lung function. We also found that the transcriptional signature of self-disclosure was also significantly associated with other measures of asthma, such as nightly asthma symptoms (r = 0.27, p=1.2 * 10$^{-5}$) and asthma severity (r = −0.32, p=2.9 * 10$^{-7}$), echoing the large body of work on the importance of self-disclosure for health (*Pennebaker, 1995*). These results provide a potential mechanism through gene expression changes in leukocytes for previously reported links between SES and asthma

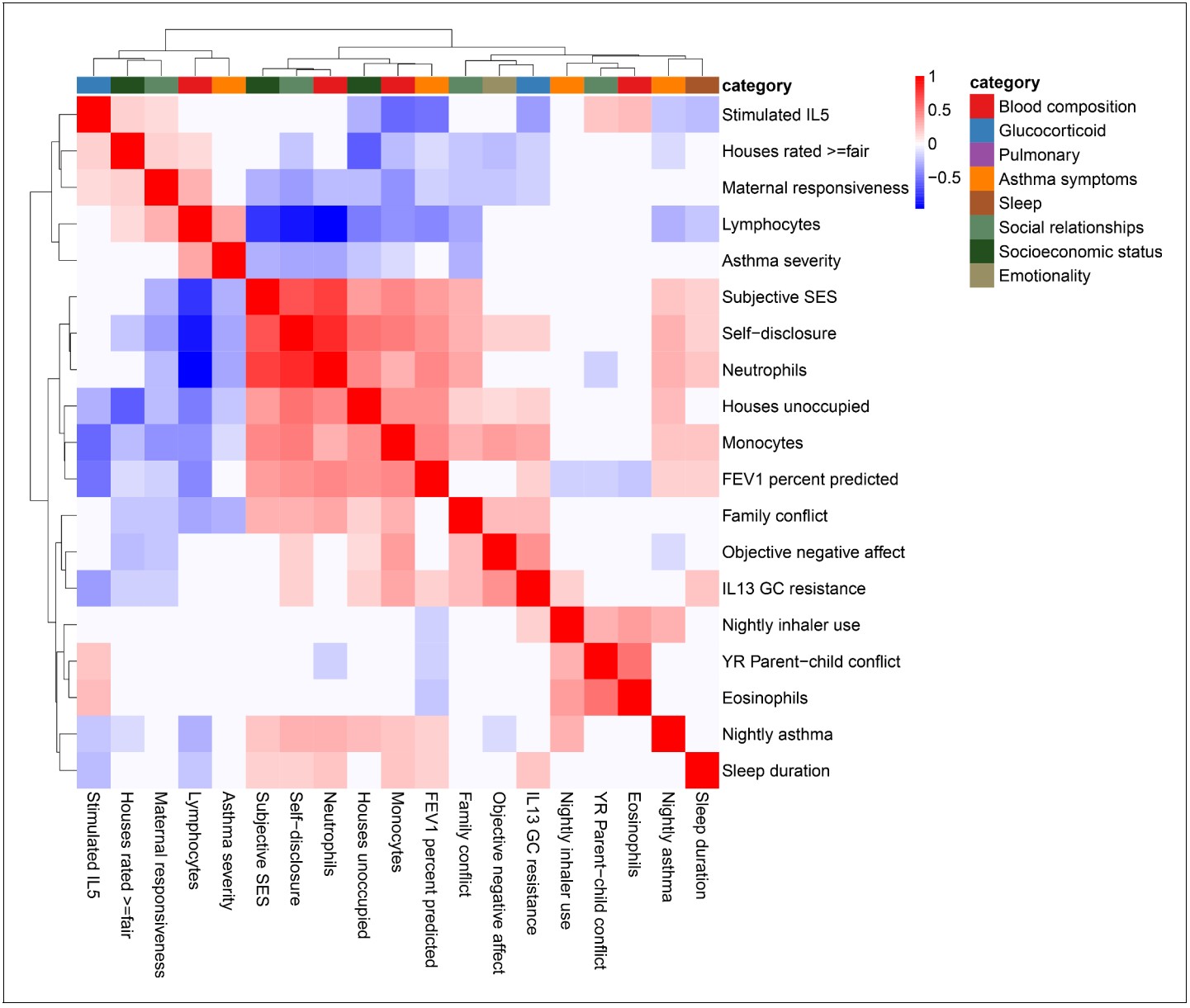

**Figure 2.** Correlation among psychosocial and clinical transcriptional signatures. Heatmap of Pearson's correlations between transcriptional signatures explaining at least 1% of variance. Heatmap color indicates strength and direction of correlation; white indicates p-value>0.05. Hierarchical clustering of variables is represented above the heatmap, with colors indicating categories for each variable as indicated in the legend.
The online version of this article includes the following figure supplement(s) for figure 2:

**Figure supplement 1.** Network representation of correlations between transcriptional signatures of psychosocial experiences (edge width reflects absolute value of Pearson's correlation score, edge color reflects positive [red] or negative [blue] correlation).

**Figure supplement 2.** Network representation of correlations between all transcriptional signatures (edge width reflects absolute value of Pearson's correlation score, edge color reflects positive [red] or negative [blue] correlation).

symptoms (*Litonjua et al., 1999*; *Mielck et al., 1996*). Past research has also found emotionality to be a strong predictor of asthma severity (*Lehrer et al., 1993*).

Notably, the transcriptional signatures of blood composition were also associated with asthma symptoms, with a positive correlation for proportion of lymphocytes and negative correlation for proportion of neutrophils (*Figure 2—figure supplement 1*). Given the important role of several blood cell types in asthma severity and exacerbations (*Mamessier et al., 2008*; *Vedel-Krogh et al., 2017*; *Bigler et al., 2017*; *Ray and Kolls, 2017*; *Lima-Matos et al., 2018*; *Casciano et al., 2016*;

*Sinz et al., 2017*), it is possible that transcriptional changes associated with blood composition mediated the correlations between psychosocial experiences and asthma outcomes.

## Genetic interactions with psychosocial factors affect gene regulation

To directly investigate whether transcriptional signatures associated with negative psychosocial experiences contribute to inter-individual variation in asthma risk, we used eQTL mapping combined with TWAS (*Gusev et al., 2016*). TWAS uses eQTLs as instrumental variables (IVs) to causally link gene expression to phenotypes. To examine local genetic effects on leukocyte gene expression, we performed cis-eQTL mapping and identified 8610 genes with at least one eQTL (eGenes, 10% false discovery rate [FDR], *Figure 3—source data 1*). These eGenes were enriched in Genotype-Tissue Expression Project (GTEx) whole blood eGenes (*Aguet et al., 2020*) (Fisher's test OR = 3.2, p-value<$2.2 * 10^{-16}$), but we also identified additional 1801 eGenes that were not detected by GTEx in whole blood.

We used the method for probabilistic TWAS analysis (PTWAS) (*Zhang et al., 2020*), which improves upon previous TWAS methods by ensuring only strong IVs are used, and is designed to allow for validating the causality assumption (see Materials and methods). We identified 2806 eGenes in the GTEx dataset that were causally associated with asthma and allergic diseases (hay fever, eczema, and allergic rhinitis) (5% FDR). Of these, 853 were eGenes in our dataset (*Figure 3— source data 3*). Here, we interrogated whether these causal genetic effects can be modulated by psychosocial factors through gene-environment interactions. To examine the genotype-by-environment effects of psychosocial experiences and blood composition on gene expression, we used the imputed transcriptional signatures for the entire cohort of 251 individuals. In addition to imputing missing data, we argue that these transcriptional signatures may better capture the environmental effects on the state of the cells at the molecular level (i.e., after denoising), compared to the observed variables. This is because observed variables have high levels of noise, and the measured values may not reflect the true biological effect. Therefore, we used the predicted values for all participants, including those for whom the variables were directly measured (denoising). This is similar to the context eQTL approach (*Zhernakova et al., 2017*) that uses other genes as a proxy variable for the environment, but here the 'context' is more easily interpretable because it is defined by a transcriptional signature associated with a specific psychosocial factor. Similarly, cell-type composition imputed from gene expression was used to map cell-type interaction QTLs for 43 cell-type-tissue combinations in the GTEx v8 dataset (*Kim-Hellmuth et al., 2020*).

For each of the eGenes identified in our dataset, we tested the lead eQTL for an interaction effect (see Materials and methods) with any of the transcriptional signatures. We discovered 344 significant interaction eQTLs across 134 unique genes (10% FDR; *Figure 3a*, *Figure 3—source data 4*). We found interaction eQTLs for all four blood composition signatures (proportion of lymphocytes, neutrophils, monocytes, and eosinophils with 81, 65, 27, and 25 GxE interactions, respectively), which represent cell-type-specific eQTLs (57.6% of all GxE eQTLs). 101 of the 108 blood-interacting eGenes (93.5%) were also identified as genes with interaction eQTLs with cell-type composition in GTEx whole blood (*Kim-Hellmuth et al., 2020*). We identified 124 GxE interaction effects on gene expression with psychosocial experiences across 77 genes, including self-disclosure (48 genes), subjective SES (40 genes), and objective maternal responsiveness (16 genes) (*Figure 3—source data 5*). These only partially overlapped (77%) GxE effects observed for blood composition and included interactions specific to psychosocial factors (*Figure 3c*). To evaluate whether the interactions with psychosocial experiences may be mediated by cell composition, we repeated the GxE mapping after correcting for blood cell composition. We observed 125 significant GxE effects (10% FDR) after removing the effect of blood composition differences (*Figure 3—source data 6*).

To validate these GxE results, we expanded the sample size for all variables with a transcriptional signature. We found that the GxE eQTLs detected with the measured variables (*Figure 3—source data 7*) were significantly enriched for low p-values in the GxE eQTLs detected with the transcriptional signatures (*Figure 3A*, *Figure 3—figure supplement 1*), and the interaction effects were highly significantly correlated (*Supplementary file 1g*).

Next, we explored the overlap between the GxE genes and previously published datasets that measured interactions with different environments (N = 134 genes, see Materials and methods and Supplementary text). We found that 94.8% of our GxE genes replicated in other datasets of GxE in gene expression (p<0.05). For example, 62 interaction eGenes for psychosocial experiences

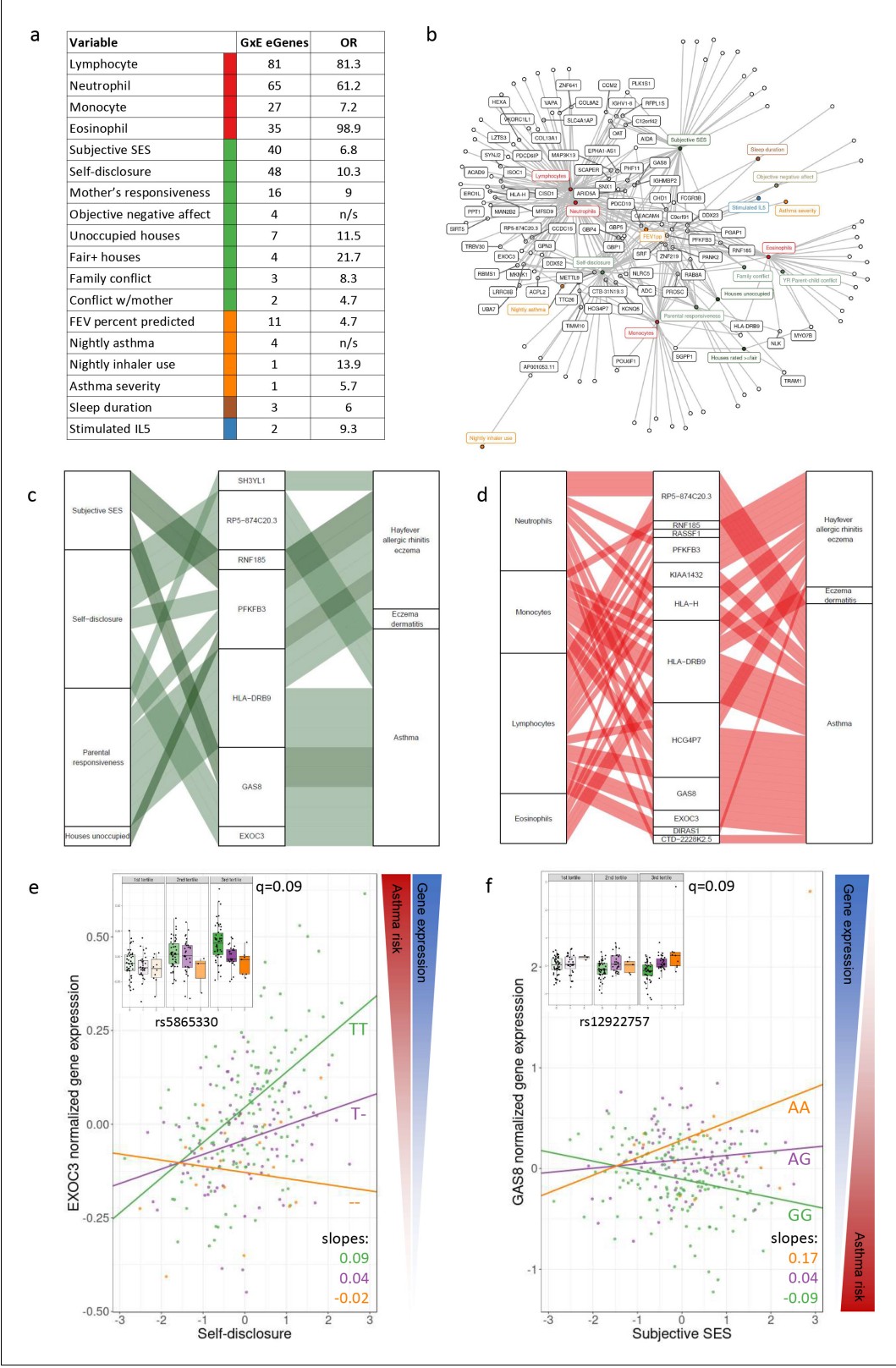

**Figure 3.** GxE effects on gene expression and asthma risk. (**a**) Interaction expression quantitative trait locus (eQTL) results. GxE genes: number of significant GxE interactions with transcriptional signatures at 10% FDR; OR: odds ratio of enrichment of GxE genes with measured variable (p<0.01) in GxE genes with transcriptional signatures (p<0.01). (**b**) Network of interactions between environments and eGenes. Each node represents an eGene with an interaction eQTL (black) or a variable that modulates the genetic effect on gene expression. Only nodes with at least two interactions are

*Figure 3 continued on next page*

*Figure 3 continued*

labeled. Edges represent significant interaction eQTLs (10% FDR). (c, d) Causal gene-complex trait interactions identified through transcriptome-wide association studies (TWAS) are modulated by psychosocial experiences. Psychosocial variables (c) or blood composition (d) are in the left column, eGenes in the central column and complex traits in the right column. A connecting line represents either a causal link between eGene and asthma or allergic disease trait identified through TWAS (middle to right) or a significant interaction eQTL (left to middle). (e, f) Examples of genes causally associated with asthma and with GxE effects that modulate genetic risk. Both genes are causally associated with asthma in TWAS. Each dot is an individual. The same data are presented in the inset and main figure within each panel. In the main figure, the trend lines represent the best model fit between the psychosocial variable and gene expression for each genotype class. The slope of each line and the q-value for the GxE effect are also reported. The boxplot in the inset represents the same normalized gene expression data across the three tertiles of the psychosocial variable.
The online version of this article includes the following source data and figure supplement(s) for figure 3:

**Source data 1.** Results of cis-expression quantitative trait locus (cis-eQTL) mapping permutation pass with FastQTL correcting for top 18 gene expression principal components [PCs] (space-delimited file).
**Source data 2.** Results of cis-expression quantitative trait locus (cis-eQTL) mapping permutation pass with FastQTL without correction for gene expression principal components [PCs] (space-delimited file).
**Source data 3.** Overlap of eGenes with psychosocial effects and significant probabilistic transcriptome-wide association studies (PTWAS) association results (5% FDR) for asthma and allergic disease.
**Source data 4.** Results of cis-interaction-expression quantitative trait locus (eQTL) mapping (tab-delimited file).
**Source data 5.** Overlap of GxE interactions and significant probabilistic transcriptome-wide association studies (PTWAS) association results (5% FDR).
**Source data 6.** Results of cis-interaction-expression quantitative trait locus (eQTL) mapping after correcting for cell composition effects (tab-delimited file).
**Source data 7.** Results of cis-interaction-expression quantitative trait locus (eQTL) mapping with measured variables (tab-delimited file).
**Figure supplement 1.** QQplots of expression quantitative trait locus (eQTL)-transcriptional signature interaction test permutation-corrected p-values, combined for blood composition (top) and psychosocial variables (bottom).
**Figure supplement 2.** Genetic variants interact with psychosocial environments to alter expression of genes linked to asthma and allergic disease.
**Figure supplement 3.** Genetic variants interact with psychosocial environments to alter expression of genes linked to asthma and allergic disease.
**Figure supplement 4.** Causal gene-complex trait interactions identified through transcriptome-wide association studies (TWAS) are modulated by psychosocial experiences.
**Figure supplement 5.** QQplots of interaction expression quantitative trait locus (eQTL) mapping test p-values (black), permutation p-values (green), and permutation-corrected p-values (pink).

overlapped with interaction eGenes in response to pathogens (*Lee et al., 2014*; *Barreiro et al., 2012*; *Çalışkan et al., 2015*; *Nédélec et al., 2016*). This result may indicate that negative psychosocial experiences lead to genotype-specific adverse health effects by influencing the same immune pathways activated by infections. Furthermore, psychosocial experiences may modify the individual response to pathogens and affect health outcomes.

## Risk for asthma is modulated by GxE

We hypothesized that genes and pathways that contribute to asthma risk are also involved in asthma symptom severity through gene regulatory variation in immune cells. We investigated whether genes associated with risk for asthma were modulated through psychosocial experiences (E) and/or GxE effects. Among the genes causally linked to asthma or allergic disease risk by PTWAS, expression of 169 genes was modulated by psychosocial environments, including self-disclosure (125 genes), subjective SES (105 genes), family conflict (30 genes), percent unoccupied houses in the neighborhood (27 genes), maternal responsiveness (17 genes), objective negative affect (e.g., feeling sad or angry, 11 genes), child-reported conflict with parent (9 genes) and percent ≥fair houses in the neighborhood (2 genes) (*Figure 3—source data 5*). The genetic effect on gene expression is modulated by psychosocial factors through GxE for seven genes causally implicated in asthma (four genes) and allergic diseases (four genes) (*Figure 3c–f*, *Figure 3—figure supplements 2–4*). For example, higher expression of the Exocyst Complex Component three gene (*EXOC3*) is associated with an increased risk of asthma. We found that self-disclosure, which is the extent to which the youths talk about their thoughts and feelings, increases expression of this gene only for individuals carrying at least one copy of the T allele at rs5865330 (*Figure 3e*). The genetic effect was even more pronounced in the highest tertile of self-disclosure (*Figure 3e*, inset). Lower expression of the Growth Arrest Specific 8 gene (*GAS8*) is associated with an increased risk of asthma. The A allele at rs12922757 increases expression of this gene only in individuals with perceived high SES, thus reducing the risk of disease (*Figure 3f*). A similar effect, and in the same direction, is found for *GAS8* and higher self-disclosure (*Figure 3—figure supplements 2* and *3*).

## Discussion

In this study, we collected a unique dataset with genome-wide gene expression paired with extensive and accurate assessment of each participant's biological and psychosocial functioning, across a variety of domains known or likely to be relevant for asthma. We developed a new approach to denoise and impute the transcriptional signatures of asthma symptoms and psychosocial experiences in peripheral blood. Longitudinal data collected on the same individuals validated the transcriptional signatures imputed on an unobserved later time point, mirroring the changes on phenotype. This demonstrates that the molecular signature of psychosocial experiences on immune cells can track changes over time and can be used to analyze cohorts where these variables are not available.

We showed overlap between transcriptional signatures of asthma symptoms and both SES and social relationships, thereby demonstrating that molecular blood gene expression pathways exist through which psychosocial experiences can affect asthma. While some of the variables used to describe pulmonary function may not directly reflect a conventional clinical endpoint, the fact that they were associated with significant transcriptional signatures indicates that they may still have pathophysiological relevance for individuals with asthma. For example, we found a significant transcriptional signature for FEV1, which is endorsed by the National Asthma Education and Prevention Program as a means for grading asthma severity (*Birnkrant, 1997*). Interestingly, we detected a higher number of significant correlations across imputed variables compared to the measured variables. This observation supports the effectiveness of the denoising procedure that we used to define transcriptional signatures.

Correlations between asthma phenotypes and psychosocial variables may be partially due to changes in blood compositions. Correlations between asthma severity and blood composition are supported by previous findings from the U-BIOPRED cohort where the number of genes differentially expressed between individuals with severe asthma and healthy controls was reduced by 90% after accounting for blood cell composition (*Bigler et al., 2017*). This is not surprising as several partially overlapping endotypes of asthma have been described to date, distinguished by pro-inflammatory contributions from different immune cell types. The most common asthma subtype is characterized by the involvement of T helper type 2 cells (Th2) sensitized primarily to allergens and subsequent eosinophilic airway inflammation triggered by the type two cytokines (particularly Il-5) (*Woodruff et al., 2009*). However, in non-allergic individuals, eosinophilic inflammation may be triggered by other immune cell types (*Brusselle et al., 2013*). Elevated levels of neutrophils have been associated with more severe asthma and suggested as an alternative mechanism to eosinophilic inflammation (*Ray and Kolls, 2017*). Further work to dissect the contributions of each cell type can be accomplished in future studies with single-cell transcriptomics.

Genes sets associated with psychosocial variables were enriched for different Gene Ontology functions and pathways. For example, self-disclosure is associated with genes distinctly enriched for neutrophil-mediated immunity (*Figure 1—figure supplement 3*), while parent-reported conflict with child is associated with expression of genes enriched for erythrocyte differentiation. This result suggests that response to negative psychosocial experiences involves processes outside of the scope of the CTRA, which was designed to capture inflammation, antibody production, and type I interferon response. Psychosocial experiences change over time because of the children's development as a result of broader changes in the urban environments and as a consequence of shifts in family dynamics. These changes of the psychosocial experiences are reflected longitudinally in the gene expression of immune cells and may modify the asthma symptoms and overall health. As we gain additional knowledge on the mechanisms connecting psychosocial experiences to disease, these results can be useful to support the need for social interventions that may ultimately lead to improved overall health. For example, family counseling may improve the psychosocial environment of children with asthma, ameliorating their symptoms, and reducing the impact of systemic health disparities. These social interventions may be implemented independently or together with drug treatments, and their impact could be further monitored through gene expression with larger longitudinal samples in future studies.

Pioneering work by our group and others has shown that environmental effects on gene expression and their interactions with genetic factors can play a very important role in regulating genes that are associated with disease (*Moyerbrailean et al., 2016*; *Richards et al., 2019*; *Findley et al., 2019*; *Knowles et al., 2017*; *Nédélec et al., 2016*; *Zhernakova et al., 2017*). This is also applicable

to asthma genetics (*Rava et al., 2015*) and exposure to rhinovirus infection (*Çalışkan et al., 2015*) and cytokines (*Thompson et al., 2020*).

Our study demonstrates that many psychosocial experiences leave an impact on gene expression, including genes that are known to be associated with asthma. When considering genetic effects on gene expression, 21% of the 8610 eQTL we discovered were not found in GTEx whole blood samples. These newly identified genetic effects may be due to limited power when performing an overlap between results from different eQTL studies. An alternative explanation is that we captured context-specific genetic effects that are due to differences between our cohort and GTEx samples in cell-type composition, ancestry, age, psychosocial environment, and/or the asthma status. Indeed, our GxE analysis identifies 344 instances of eGenes with context-specific genetic regulation of gene expression, including 124 instances of GxE with psychosocial experiences.

One outstanding challenge in human complex trait genetics focuses on the portability of polygenic risk scores across population groups or environments (*Mostafavi et al., 2020*). The 124 instances of GxE with psychosocial experiences are particularly relevant when evaluating a polygenic risk score for asthma phenotypes. For example, the individual contribution of a gene can be modified by SES in one direction and in a different direction for a different gene.

Here, we show that these altered gene expression immune profiles may in turn exacerbate asthma symptoms in children living in inner cities, who are exposed to riskier psychosocial environments. Using human genetics tools, we established that psychosocial factors can modulate the causal genetic effects between gene expression and asthma. Importantly, our results demonstrate that psychosocial factors, such as self-disclosure and SES, modulate genetic risk of asthma and other allergic diseases through altered peripheral blood gene expression.

# Materials and methods

## Study participants

Participants were included from an ongoing longitudinal study, ALOFT ( recruited from November 2010 to July 2018, Wayne State University Institutional Review Board approval #0412110B3F). The ALOFT study investigates the links between family dynamics, biological changes, and asthma morbidity among youth from the Detroit metropolitan area. Participants were recruited from local area hospitals and schools (for recruitment details, see Supplemental text). To be included in the study, youth were required to be between 10 and 15 years of age at the time of recruitment and diagnosed with at least mild to persistent asthma by a physician (with diagnosis confirmed from medical records). Youth were screened for medical conditions and medications that might affect asthma and associated biological markers. Only one participant reported current oral corticosteroid use. The full sample included 297 youth and their primary caregivers (typically mothers, referred to as 'parent' below). However, only youth with valid gene expression data were included in this investigation. Thus, the sample comprised 251 youth (148 boys and 103 girls), whose average age was 12.89 years old (*sd* = 1.77 years), and at least one parent. Psychosocial and biological variables, including asthma measures, were available for a subset of 119 participants. For a subset of up to 103 participants, we have collected longitudinal data (either 1- or 2 year follow-up), which we used to validate the transcriptional signatures. For cross-sectional and longitudinal sample sizes for each variable, refer to *Supplementary file 1b*.

## Participant recruitment and collection of psychosocial and biological variables

The parent completed a telephone screening interview to determine eligibility in the study. Written assent and consent were obtained from the participating youth and their parent, respectively.

### In-lab assessments

The participating youth and parent visited the laboratory, where they completed background questionnaires on a computer and individual interviews assessing stress and asthma management. The parent reported demographics, including their annual income and education level, and completed measures of subjective SES (the McArthur ladder [*Adler et al., 2000*], adjusted so that 1 corresponds to the lowest status and 10 to the highest), neighborhood stress (*Ewart and Suchday,*

*2002*), conflict with their child (the Parental Environment Questionnaire; *Elkins et al., 1997*), and depressive symptoms (the CES-D; *Radloff, 1977*). The zip code for each family was also collected and used to retrieve objective measures of neighborhood quality based on census block data from 2010 and Data Driven Detroit (collected in 2009), including the percentage of houses rated as fair in quality or better, percentage of houses currently unoccupied, and the percent of people in that area living below the poverty line (*Zilioli et al., 2017*).

At the same time, youth reported on demographics, warmth received from their mother (Parental Behavior Inventory; *Schaefer, 1965*), conflict with their mother, the quality of their family environment (the Risky Families Questionnaire; *Felitti et al., 1998*; *Taylor et al., 2004*), depressive symptoms (the Child Depression Inventory; *Kovacs, 2011*), and the frequency and severity of their asthma symptoms (the Teen Asthma History). Youth also reported on their parents smoking inside the household. However, due to low prevalence as well as uncertainty on whether the parents were present in the household during the 4 days of data collection, we decided to not use this information in our analyses. They also completed a spirometry test using the nSpire Health KoKo PFT to obtain the following pulmonary measures: FEV1 percent predicted, FVC percent predicted, FEV1/FVC percent predicted. Also at this visit, the youth and parent were given detailed instructions regarding a 4-day daily assessment period. The laboratory visit lasted approximately 2 hr.

## In-home assessments

For 4 days following the laboratory visit (two weekdays and two weekend days), youth and their parent completed daily assessments. Both youth and their parent completed daily diaries each evening about their experiences throughout their day, and sleep diaries each morning about the quality of their sleep. Daily diaries contained items assessing their positive (i.e., happy, interested, excited, and proud) and negative (i.e., sad, angry, upset, worried, distressed) affect, and how much affection and conflict they witnessed between their parents. Youth were also asked to think about the most important and meaningful conversation they had with someone that day and the extent to which they talked about their thoughts and feelings during that conversation (to measure self-disclosure), and how understanding, validating, and caring their conversation partner was (to measure perceived responsiveness). The sleep diaries contained the Pittsburgh Sleep Scale (*Monk et al., 1994*), which assesses sleep latency (how long to fall asleep), sleep efficiency (how much time in bed spent sleeping), the number of awakenings throughout the night, the total duration of sleep in hours, and the quality of the sleep. Through the daily and sleep diaries, the participants provided information on the following measures of asthma: severity and frequency of daily and nightly asthma symptoms (wheezing, shortness of breath, coughing, chest tightness, other) and nightly inhaler use. Description of daily diary and sleep diary items used in this investigation is included in *Supplementary file 2*. When youth completed the daily and sleep diaries (i.e., at awakening and before bed), they used a peak flow meter twice to measure peak flow, with the best score between the two assessments used as our measure of morning and evening peak flow. Only daily and sleep diary reports from youth are used in this investigation. Additionally, youth provided four samples of saliva daily for 4 days at wakeup, 30 min after wakeup, before dinner, and immediately before bed using passive drool methods. Sample time was recorded by participant report, time stamps, and MEMS 6 TrackCap monitors (Aardex Ltd., Switzerland). Samples were initially stored in participants' refrigerators, but upon return to the lab, saliva samples were stored in the laboratory refrigerator at −20°C until assayed. To reduce positive skewness, we natural log transformed the cortisol values (raw cortisol +1). Hierarchical linear models were run in HLM to extract the average diurnal cortisol intercept, slope, and cortisol awakening response (CAR) for each participant. Finally, participants wore the Electronically Activated Recorder (EAR) in their front pocket or in a belt clip provided from the time they woke up until bedtime. The EAR captured 50 s of sound every 9 min (*Mehl et al., 2001*). EAR data were coded by trained coders using the Everyday Child Home Observation (ECHO) coding system (*Tobin et al., 2015*). Specifically, for this investigation, we use codes of wheezing, positive affect (i.e., happy, interested, excited), negative affect (i.e., sadness, anger, upset, worry, distress), maternal responsiveness (i.e., how much the mother expresses pride, support, and warmth towards the youth), and family conflict (i.e., whether an argument, conflict, fight, or yelling was overheard). Scores for each EAR-observed behavior reflect a mean of the total recordings in which the behavior

was observed during waking hours. After completion of the in-home assessment period, the participants returned study materials and the EAR. Youth and parents were compensated for their time.

Additional details on the measures collected in-home are provided in *Supplementary file 2*. Descriptive and reliability statistics can be found in *Supplementary file 1b*. Correlations between measures can be found in *Figure 1—figure supplement 1*.

### Biological sample collection

Following the daily assessment period, a peripheral blood draw was conducted for each youth participant. Each youth provided 16, 4, and 8 ml of peripheral blood collected into Vacutainer Cell Preparation Tubes (Becton Dickinson and Co., East Rutherford, NJ) for PBMC (FICOLL gradient vacutainers), DNA (sodium citrate vacutainer, Fisher Scientific catalog #BD-366415), and RNA (EDTA vacutainer) extraction, respectively. Peripheral blood mononuclear cells (PBMCs) were extracted from this sample, as previously described (*Weckle et al., 2015*). All PBMC samples were phenotyped for glucocorticoid (GC) resistance in an established in vitro assay (*Marin et al., 2009*) measuring the levels of IL-5, IL-13, and IFN-$\gamma$ in the supernatant (Quantikine ELISA D5000B, D1300B, and DIF-50, R&D Systems, Minneapolis, MN). Specifically, PBMCs cultured in RPMI-1640 solution (Life Technologies, Carlsbad, CA) supplemented with 10% FBS (Life Technologies) and 2% HEPES (Sigma-Aldrich, St. Louis, MO) were stimulated for 48 hr with PMA + ionomycin (phorbolmyristate acetate 25 ng/ml, Fisher Scientific, Hanover, IL; ionomycin calcium salt, 1 μg/ml, Sigma-Aldrich) and treated with hydrocortisone (28 nmol/l, Sigma-Aldrich) or vehicle control. GC resistance was calculated as log-fold change of cytokine level in hydrocortisone condition over control and averaged over two replicates. DNA was extracted using DNA Blood Mini Kit (Qiagen, Germantown, MD), and RNA was extracted using LeukoLOCK Total RNA Isolation System (Thermo Fisher Scientific, Waltham, MA).

Each of the aforementioned measures were collected annually for a period of 2 years (three data collection waves) from participants who provided continued informed consent. In this study, we used both cytokine levels in stimulated PBMCs (e.g., IL5 stimulated), as well as log-fold change in cytokine levels between stimulated and stimulated + glucocorticoid-treated condition (e.g., IL13 GC resistance).

## Genotype data

All individuals in this study were genotyped from low-coverage (~0.4×) whole-genome sequencing and imputed to 37.5M variants using the 1000 Genomes database by Gencove (New York, NY). The genotype accuracy at this sequencing depth has 98.22% positive agreement and 99.82% negative agreement compared to genotyping arrays (*Wasik et al., 2021*; *Li et al., 2021*). These data were also used for sample quality control (see Ancestry QC, Sex QC and Genotype QC) and to calculate the top three principal components [PCs] to use as covariates in all statistical analyses.

## Genotype QC

To detect potential sample swaps that may have occurred in sample processing or library preparation, we compared genotypes of RNA and DNA samples from all individuals. We used samtools mpileup function to obtain genotypes from each individual's RNA-seq bam files for NCBI dbSNP Build 144 variants and kept only variants with more than 40 reads coverage. We used bcftools gtcheck function to compare genotype calls across all biallelic SNPs in all DNA and RNA samples. RNA samples that failed to cluster with their respective DNA sample were repeated (library preparation and sequencing). If the discrepancy was not resolved, these samples were excluded from the analysis. A total of 251 samples passed this QC filter. Ultimately, the pairwise discordance rate between genotype calls from RNA and their respective DNA samples from the same individual ranged between 0.03 and 0.12. In contrast, the pairwise discordance rate between all the other unrelated samples ranged from 0.20 to 0.33. This rate is not a direct measure of genotype call accuracy, but it is useful to identify possible sample swaps and mismatches.

We also used the DNA-derived genotype information to confirm none of the participants were related. We performed identity-by-descent (IBD) analysis by maximum likelihood estimation (MLE) using the R package SNPRelate (version 1.16.0). As input we used random 1500 SNPs passing the

following criteria: minor allele frequency (MAF) >0.05, missing rate <0.05, LD threshold <0.2 (*Figure 1—figure supplement 4*).

## Ancestry and sex QC

For the individuals for whom the data was available, we plotted self-reported ethnicity against percent global African ancestry defined as the sum of West, East, Central, and North African global genetic ancestries calculated by Gencove (*Figure 1—figure supplement 5*). All samples were in agreement with self-reported ethnicity. Three participants who identified as multiracial were found to be of admixed African and European ancestry based on genotype analysis provided by Gencove. To check consistency of self-reported sex against genetic data, we plotted fraction of reads mapping to the Y chromosome for all samples. We noted a clear separation between the sexes with no outliers (*Figure 1—figure supplement 6*).

## RNA-seq data collection and preprocessing

Total RNA was extracted using LeukoLOCK (Thermo Fisher), which captures total RNA from neutrophils, eosinophils, basophils, monocytes, and lymphocytes, and preserved at −80°C. All RNA samples had a RNA Integrity Number (RIN) of at least 6 measured on Agilent Bioanalyzer. Library preparation was performed in batches of up to 96 samples (with multiple samples from the same participant always processed within the same batch) on 1–4 µg total RNA, per standard Illumina Tru-Seq Stranded mRNA library preparation protocol, and sequenced on Illumina NextSeq500 to a depth of 21 million (M) to 76M reads, mean 41M reads (150 bp paired-end). HISAT2 (*Kim et al., 2015*) was used to align demultiplexed reads to the human genome version 'GRCh37_snp_tran', which considers splicing and common genetic variants. Aligned and cleaned (deduplicated) reads were counted using HTSeq and GRCh37.75 transcriptome assembly across 63,677 genes. Postsequencing quality control included removal of samples with excess PCR duplicate rate (>60%) and genotype QC check against respective DNA sample. For all gene expression analyses, genes on sex chromosomes and genes with expression below 6 reads or 0.1 counts per million in at least 20% of samples were dropped. The final RNA-seq dataset consists of 251 unique samples and 18,904 genes.

## Differential gene expression analysis

We used DESeq2 v1.22.1 (*Love et al., 2014*) to test for differential gene expression across the 23 psychosocial experiences using a likelihood ratio test (LRT) in 119 individuals from the first wave of data collection. To adjust for potential confounders, we included as covariates the three top PCs of a matrix of possible confounders that included RIN, site of RNA extraction, library preparation batch, percent reads mapping to exons, percent non-duplicate reads, age, sex, height, weight, top three genotype PCs, and the four transcriptional signatures of blood composition (*Figure 1—figure supplement 7*), except when testing for the effect of blood composition differences on gene expression where we have not included transcriptional signatures of blood composition when calculating PCs of potential confounders. Many of these confounders are very correlated, and the three top PCs explained 99.7% of their variance. *Supplementary file 1h* represents correlations between individual covariates and the three top PCs of the covariate matrix. In short, PC1–PC3 largely represent weight, height, and age, respectively. For each tested variable, the LRT is then used to compare between two models: GE ~ cvPC1+cvPC2+cvPC3+tested_variable (full model) and GE ~ cvPC1+cvPC2 +cvPC3 (reduced model). To control for FDR, we used the default independent filtering step and multiple test correction implemented in DESeq2. *Supplementary file 1e* lists differentially expressed genes at 10% FDR, while *Supplementary file 1f* contains the full results of the analysis. *Supplementary file 1i* lists differentially expressed genes for blood composition measures at 10% FDR, while *Supplementary file 1j* contains full DESeq results for this analysis.

## GO and pathway enrichment analyses

We used the R package clusterProfiler (*Yu et al., 2012*) to run GO, KEGG, and REACTOME enrichment analyses (hypergeometric test) across genes upregulated and downregulated compared to the background of all expressed genes (*Figure 1—figure supplement 3*). Enriched categories were defined at 5% FDR.

## Imputation and denoising of transcriptional signatures

We assume that the observed variables have high levels of noise and the measured values do not reflect the true biological effects. Therefore, we used the predicted values for all participants, including those for whom the variables were directly measured (denoising). We developed an approach to impute and denoise a transcriptional signature for psychosocial, environmental, and other phenotypic variables based on Generalized Linear Models with Elastic-Net Regularization. First, we normalized the count data using the voom function in the *limma* v3.38.3 package in R (*Law et al., 2014*). Second, we regressed out the following confounding factors: RIN, percent reads mapping to exons, percent non-duplicate reads, site of RNA extraction, library preparation batch, sample collection wave, age, sex, height, weight, genotype PC1, genotype PC2, and genotype PC3. Third, we used the R package *glmnet* v2.0–16 in R/3.5.2 (Gaussian model), with a relaxed alpha = 0.1 to allow for highly co-regulated genes to be included in the prediction model.

The predicted values are imputed based on the generalized linear models with penalized maximum likelihood built using *glmnet* for each variable separately, according to the general model:

$$\text{Phenotype or Environment} = \text{intercept} + \beta_1 \text{E(gene1)} + \beta_2 \text{E(gene2)} + \ldots + \beta_n \text{E(genen)}, \qquad (1)$$

where E (gene$_n$) is normalized expression of gene n, $\beta_n$ is its estimated coefficient, and n is minimized via penalized maximum likelihood with elastic-net mixing parameter $\alpha$ set to 0.1 (0 representing ridge regression, 1 representing lasso regression).

The best fit for the model was used to predict, for both observed and unobserved samples, the biological impact on gene expression of the relevant variable. We did this because the measurement error for the observed variables will also not be uniform as some individuals may not respond accurately or truthfully (e.g., to self-disclosure questions) to all questions. If the objective was to estimate the measurement including its biases and errors, the 'technical' variance for the observed variable would be smaller if we were to use the observed values. However, we use the denoised/imputed values, where the 'biological' variance, or the error between the fitted value of the signature with respect to the true unobserved biological impact, would be the same for both the measured and non-measured individuals. Overall, our procedure should not create any biases but rather decrease the variability of the imputed/denoised variables, thus reducing the chance of false-positive GxE.

Leave-one-out cross-validation was used to evaluate the best models. We used the cross-validated mean square error (MSE) metric and its standard deviation to evaluate which signatures were more predictive. We calculated the $R^2$ for each of the models based on the % MSE reduction from cross-validation. To compare the results that would be achievable with the CTRA-based approach, we used the same method but we limited the molecular signature to only include the 53 genes that are used to calculate the CTRA score (*Fredrickson et al., 2013*). 48 of the 53 genes comprising the CTRA are measurable in our sample (CTRA genes below detection: IL1A, IFIT1L, IFITM5, IFNB1, IGLL3). We compared the fraction of variance explained between the CTRA-based and unrestricted models (*Figure 1—figure supplement 8*).

## Correlation between transcriptional signatures

We used Pearson's correlations to evaluate overlap between transcriptional signatures of variables explaining at least 1% of variance imputed on the entire cohort of 251 participants as in *Equation (1)*. We only considered correlations with p-value<0.05.

## Longitudinal replication

We collected a second time point (approximately 1 or 2 years after the time point used in current analyses) for a subset of 13 variables – subjective SES, self-disclosure, YR parent-child conflict, stimulated IL5, IL13 GC resistance, eosinophils, lymphocytes, monocytes, neutrophils, FEV1 percent predicted, nightly asthma symptoms, nightly inhaler use, and asthma severity – to validate the transcriptional signatures. We considered the longitudinal changes in the transcriptional signatures imputed from the new gene expression data and compared them to the changes in the observed variable between the two time points. Note that the transcriptional signature is imputed for the second time point from gene expression samples that are not included in the training set. We used Spearman's correlation to compare the changes from the imputed transcriptional signature to those directly observed.

## cis-eQTL mapping

We calculated gene expression residuals (as in the imputation and denoising approach) and then used FastQTL (*Ongen et al., 2016*) with adaptive permutations (1000-10,000). For each gene, we tested all genetic variants within 1 Mb of the transcription start site (TSS) and with cohort MAF > 0.1, for a total of 17,679 genes and 82,679,170 variant-gene pairs tested. We optimized the number of gene expression PCs in the model to maximize the number of eGenes. The model that yielded the largest number of eGenes included 18 gene expression PCs (*Figure 3—source data 1*).

## Interaction eQTL mapping

To identify interaction eQTLs, we considered the lead eQTL for each of the eGenes identified at 10% FDR by FastQTL (without correcting for gene expression principal components). This is similar to what was done by GTEx (*Kim-Hellmuth et al., 2020*) and others (*Alasoo et al., 2018*; *Kim-Hellmuth et al., 2017*), and equivalent to a very conservative pruning of all SNPs in the entire cis-association region. We did not correct for gene expression principal components because some of them are correlated with cell composition and the environmental variables, thus complicating the interpretation of the linear model. To reduce impact of potential outliers, we quantile-normalized each transcriptional signature prior to GxE testing. We fit a linear model that includes both the genotype dosage and the marginal environmental effect as well as their interaction: Expression ~dosage + transcriptional signature +dosage*transcriptional signature. To fit this model, we used the lm function in R-3.5.2. We generated an empirical null distribution of 100 million permuted p-values to correct the interaction p-values (*Figure 3—figure supplement 4*). The empirical null distribution was obtained through multiple runs of the model for each tested transcriptional signature-gene pair while permuting the genotype dosages. Storey's q-value method to control for FDR was applied on the permutation-corrected p-values for all tests within each transcriptional signature separately.

To ensure the signal detected was not solely due to cell composition differences, we repeated the GxE eQTL mapping procedure as above, while correcting for four signatures of cell composition using the following model: Expression ~ eosinophils + leukocytes+monocytes + neutrophils+dossage + transcriptional signature +dosage*transcriptional signature. *Figure 3—source data 6* contains full results of this analysis.

Interaction eQTL mapping with measured variables was performed the same way as with transcriptional signatures, except sample size was limited by the available data (*Figure 3—source data 7*).

## Replication analysis of GxE

We calculated the enrichment of GxE genes from the measured variables (p-value<0.01) in the set of GxE genes from the transcriptional signatures (p-value<0.01). To this end, we performed Fisher's exact tests on 2 × 2 contingency tables indicating whether a gene had a GxE eQTL with the measured variable (yes/no) and with the corresponding transcriptional signature (yes/no). Additionally, we calculated the correlation of the standardized interaction effect size (z-score) for each gene obtained when considering measured variables and corresponding transcriptional signatures.

To validate our GxE results, we considered the following GxE studies for which full interaction testing results are available (*Barreiro et al., 2012*; *Lee et al., 2014*; *Çalışkan et al., 2015*; *Nédélec et al., 2016*; *Moyerbrailean et al., 2016*; *Kim-Hellmuth et al., 2020*). We show numbers of our GxE eGenes (FDR < 10%) that replicated in other studies (p-value<0.05).

## TWAS analyses

To directly investigate whether discovered effects on gene expression and GxE interactions may contribute to asthma, allergic disease risk, and/or behavioral phenotypes, we used PTWAS results (*Zhang et al., 2020*) (5% FDR) as an independent source of evidence of causality between gene expression levels and asthma/allergic disease risk. PTWAS utilizes probabilistic eQTL annotations derived from multivariant Bayesian fine-mapping analysis conferring higher power to detect TWAS associations than existing methods. The evidence for causality from PTWAS is strong for the following reasons: (1) we use only strong IVs by combining the strength of multiple independent strong eQTLs for each gene and combining information across all tissues; (2) within the PTWAS

framework, we can then validate the causality assumption for each gene-trait-tissue combination. We found that the exclusion restriction criterion was violated (heterogeneity of independent estimates across multiple strong eQTLs, $I^2$ statistic >0.5) in only 0.36% of the gene-trait pairs for which we computed this statistic, none of which overlap our reported results. Using eQTL data across 49 tissues from GTEx v8, we used PTWAS to analyze GWAS summary statistics from several large-scale projects. Here, we specifically focused on the following asthma studies: GABRIEL-Asthma, TAGC-Asthma-EUR, UKB-20002–1111-self-reported-asthma, UKB-6152–8-diagnosed-by-doctor-Asthma, and allergic disease studies: EAGLE-Eczema, UKB-20002–1452-self-reported-eczema-or-dermatitis, UKB-6152–9-diagnosed-by-doctor-Hayfever-allergic-rhinitis-or-eczema. Additionally we considered other phenotypes that may be relevant for our cohort: chronotype (Jones-et-al-2016-Chronotype, UKB-1180-Morning-or-evening-person-chronotype), sleep duration (Jones-et-al-2016-SleepDuration, UKB-1160-Sleep-duration), and depressive symptoms (SSGAC-Depressive-Symptoms). To identify eGenes in children with asthma that are causally associated with asthma, we considered all 4943 eGenes that were used for the interaction eQTL analysis with a significant (10% FDR) marginal effect of the psychosocial experiences from the linear model that includes both the genotype dosage and the marginal environmental effect as well as their interaction: Expression ~ dosage + transcriptional signature +dosage*transcriptional signature.

## Acknowledgements

We thank Luis Barreiro and Noah Snyder-Mackler for comments on an earlier version of this manuscript, and members of the Luca, Pique-Regi, Zilioli, and Slatcher labs for helpful discussions. We thank the study participants and their families for taking part in this study. We thank two anonymous reviewers for their insightful comments.

## Additional information

### Funding

| Funder | Grant reference number | Author |
|---|---|---|
| National Heart, Lung, and Blood Institute | R01HL114097 | Samuele Zilioli Richard B Slatcher |

The funders had no role in study design, data collection and interpretation, or the decision to submit the work for publication.

### Author contributions

Justyna A Resztak, Data curation, Formal analysis, Validation, Investigation, Visualization, Writing - original draft, Writing - review and editing; Allison K Farrell, Data curation, Formal analysis, Writing - original draft, Writing - review and editing; Henriette Mair-Meijers, Adnan Alazizi, Data curation; Xiaoquan Wen, Resources, Methodology, Writing - review and editing; Derek E Wildman, Resources, Data curation, Supervision, Writing - review and editing; Samuele Zilioli, Resources, Data curation, Supervision, Project administration, Writing - review and editing; Richard B Slatcher, Conceptualization, Resources, Data curation, Supervision, Funding acquisition, Project administration, Writing - review and editing; Roger Pique-Regi, Francesca Luca, Conceptualization, Resources, Data curation, Supervision, Funding acquisition, Investigation, Methodology, Writing - original draft, Project administration, Writing - review and editing

### Author ORCIDs

Justyna A Resztak https://orcid.org/0000-0002-9567-7369
Roger Pique-Regi https://orcid.org/0000-0002-1262-2275
Francesca Luca https://orcid.org/0000-0001-8252-9052

## Ethics

Human subjects: Participants were included from an ongoing longitudinal study, Asthma in the Lives of Families Today (ALOFT; recruited from November 2010-July 2018, Wayne State University Institutional Review Board approval #0412110B3F).

## Decision letter and Author response

Decision letter https://doi.org/10.7554/eLife.63852.sa1

Author response https://doi.org/10.7554/eLife.63852.sa2

# Additional files

## Supplementary files

• Supplementary file 1. Supplementary tables. (**a**) Basic demographic information on 119 individuals used to train the transcriptional signature models ('training') and the entire cohort of 251 participants ('total'). Reported are the count (N) and percentage of non-missing values per each category. For each variable in parentheses, we reported the p-values for significant differences in variable distribution between the training group and the group not included in the model. (**b**) List of variables collected for the current study. DD: daily diary; SD: sleep diary; EAR: coded from Electronically Activated Recorder; YR: youth reported; PR: parent reported; CD: census data; GC: glucocorticoid; SD: standard deviation; α: Chronbach's alpha measuring reliability as the average correlation between scale items, as a function of the number of items included in the scale, N initial – sample size used for differential gene expression analysis and building transcriptional signatures (mean, SD, and α are reported for this subset), N longitudinal – number of samples with measurements from two time points available, N expanded – final sample size used for transcriptional signature and GxE validations. (**c**) Evaluation of transcriptional signatures derived using elastic net regression. For each variable, we report Pearson's correlation coefficient, p-value, cross-validated percent variance explained, and sample size for the training dataset (columns 2–5), and Pearson's correlation coefficient, p-value, and sample size for the entire cohort (columns 6–8). (**d**) Longitudinal replication of transcriptional signatures. For each variable, we report Spearman's correlation coefficient between longitudinal change in observed variable and change in the transcriptional signature, p-value, coefficient of variation of longitudinal change in observed variable, and sample size. (**e**) Differentially expressed genes (DEGs) for all psychosocial variables (10% FDR; N: sample size). (**f**) Full DESeq results for differential gene expression analyses for psychosocial variables. (**g**) Correlations between z-scores of GxE interaction effects measured using transcriptional signatures and observed data. For each variable, we report Spearman's correlation coefficient and its p-value, and odds ratio and p-value from Fisher's test for enrichment of nominally significant results (permutation-corrected p-value<0.01) between GxE expression quantitative trait locus (eQTL) testing using transcriptional signatures and observed data. (**h**) Correlations between the top three principal components of covariate matrix and individual covariates (Pearson's product-moment correlations with numeric variables, polyserial correlations with bivariate variables; ns: correlation p-value>0.05). (**i**) DEGs associated with blood cell composition (10% FDR; N: sample size). (**j**) Full DESeq results for differential gene expression analyses for blood composition variables.

• Supplementary file 2. Detailed descriptions of methods for psychosocial data collection.

• Transparent reporting form

• Reporting standard 1. STREGA reporting recommendations, extended from STROBE Statement.

## Data availability

The data are available on dbGAP (accession number: phs002182.v1.p1).

The following dataset was generated:

| Author(s) | Year | Dataset title | Dataset URL | Database and Identifier |
|---|---|---|---|---|
| Zilioli S, Slatcher RB, | 2021 | Asthma in the Lives of Families | https://www.ncbi.nlm. | dbGAP, phs002182. |

| Pique-Regi R, Luca F | Today (ALOFT) | nih.gov/projects/gap/cgi-bin/study.cgi?study_id=phs002182.v1.p1 | v1.p1 |

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

## Appendix 1

### Participant recruitment methods

We used the following three recruitment methods, in order of most-used to least-used:

1. Dr. Secord, a co-I on the study, identifies patients who meet some of the basic criteria for the study who are then sent informational letters to their home address. The letter informs them of the basic information about the study and a telephone number to contact if interested.
2. A nurse from DMC Children's Hospital approaches patients and their guardians in asthma clinic or ER and informs them about the study. If interested, patient's guardian is later contacted to further inform them about the study and screen them for eligibility.
3. Through informational flyer, posted in local asthma clinic, distributed to local K-12 schools with basic information about the study and eligibility criteria, and study coordinator contact information.

Basic information provided to potential participants across the three methods informs them that this is a family asthma study looking at everyday life and how health is affected, that there is a home or office/lab visit focused on surveys, a brief interview, and a 4-day at-home period and a follow-up where blood is collected.

### Participant eligibility

1. Child is between 10 and 15 years old.
2. Child has an asthma diagnosis of at least mild to persistent asthma.
3. At least one parent/guardian is willing to participate as well as the child.
4. Participating parent/guardian should be living with the child consistently for the last six months.
5. If unsure of which parent/guardian should participate, then the one that would have the most knowledge about health and day-to-day life.
6. If multiple children in the house have asthma and are in the study age range, the family can choose which child will participate based on who would be most agreeable to the various measures taken throughout the study.

### Identification of gene expression confounders

There are many sources of technical and biological noise in RNA-seq data. To accurately estimate gene expression differences between meaningful groups, one must account for additional sources of gene expression variation that are not of interest. To investigate the major sources of variation in the RNA-seq dataset, we performed principal component analysis (PCA) on normalized RNA-seq data and investigated (1) the correlation between first 15 PCs and each of the covariates (data not shown), and (2) proportion of variance explained by each covariate overall (*Figure 1—figure supplement 7a*), and per each gene (*Figure 1—figure supplement 7b*). The following covariates were significantly correlated with at least one PC: library preparation batch (confounded with library sequencing batch), RNA quality score (RIN), laboratory where the RNA sample was extracted, age, sex, height, weight, and ancestry (measured as first three principal components on the genotype matrix). These variables were also correlated with each other; therefore, we used PCA to calculate the PCs of all these covariates that explained >99.5% variance and incorporated them in the gene expression analyses as described in the Materials and methods.

### Blood composition has a profound effect on gene expression

Blood composition is expected to have a strong effect on overall blood gene expression as blood is a heterogeneous tissue and different cell types may contribute different transcripts to the overall gene expression. The results of differential gene expression analysis with DESeq2 using LRT are displayed in *Supplementary file 1i, j*. As expected, the number of differentially expressed genes is larger for cell types that constitute a bigger fraction of the cell pool. Accordingly, we did not find genes differentially expressed for basophils, a cell type that constitutes less than 1% of total

leukocytes. Based on these results, cell composition was accounted for in differential gene expression analyses of psychosocial variables and other analyses that distinguished between the two types of effects (mediation analysis).

## Overlap of transcriptional signatures

We investigated whether transcriptional signatures for different variables were correlated with each other, which would suggest shared transcriptional pathways for these phenotypes and environments (*Figure 2—figure supplement 1*). Transcriptional signatures of the socioeconomic measures showed strong overlap: specifically, houses rated ≥fair, unoccupied houses, and subjective SES were inter-correlated. There was also some overlap in the social relationships category, with transcriptional signatures of daily youth-reported self-disclosure and objective maternal responsiveness significantly correlated. However, we also saw correlations crossing all three variable categories. For example, subjective SES was significantly correlated with objective maternal responsiveness, family conflict, and self-disclosure; objective negative affect was correlated with self-disclosure, family conflict, objective maternal responsiveness, and houses rated ≥fair.

We were able to identify transcriptional signatures that explained at least 1% of variance for four of the five blood composition variables and two of the seven glucocorticoid variables. Models predicting blood composition performed best with 47–150 genes, and models predicting glucocorticoid variables contained 17–100 genes. Notably, the transcriptional signatures explained a far greater percent of variation in blood composition than other categories of variables. Transcriptional signatures within each category were highly correlated, showing greater overlap within all glucocorticoid response variables and all blood composition variables (except for eosinophils, which were uncorrelated with all other blood composition signatures) than seen in asthma severity or social relationship categories. When we correlated glucocorticoid and blood composition transcriptional signatures with transcriptional signatures for psychosocial experiences and asthma severity, we observed some overlap for glucocorticoid response: specifically, stimulated levels of IL-5 were associated with percent-predicted FEV1 (r = −0.51, p<0.001), unoccupied houses (r = −0.27, p=<0.001), and youth-reported parent-child conflict (r = 0.18, p=0.003). There was much more overlap in blood composition transcriptional signatures with asthma severity and psychosocial experience signatures. For example, transcriptional signature of monocytes was correlated with a large number of transcriptional signatures of both asthma and psychosocial experiences, including nightly asthma symptoms (r = 0.21, p<0.001), and FEV (r = 0.45, p<0.001), self-disclosure (r = 0.49, p<0.001), unoccupied houses (r = 0.39, p=<0.001), subjective SES (r = 0.43, p<0.001), and objective negative affect (r = 0.35, p<0.001). Lymphocytes showed a very similar pattern of correlations to neutrophils, but in the opposite direction.

## Targeted vs. unbiased approach to identifying effects of psychosocial experiences on gene expression

Previous research analyzing blood gene expression developed the CTRA (*Fredrickson et al., 2013*), which proposes expression of 53 immune genes as a composite indicator of immune system response to adverse social environments (reviewed in *Cole, 2014*). We compared the performance of our unbiased prediction model with a model limited to the 53 CTRA genes (*Figure 1—figure supplement 8*). The CTRA-based model performs better at predicting the fraction of two of the five blood cell types. This may reflect differential expression of CTRA genes in some cell types. CTRA-based and unbiased models perform similarly on psychosocial and neighborhood measures, with 9/16 traits (56%) predicted more accurately by the unbiased model. CTRA-based model performs poorly at predicting asthma and glucocorticoid phenotypes, and is outperformed by the unbiased model in 10/14 traits (71%), which is expected because it was not originally developed to capture these phenotypes. In summary, limiting the scope of the prediction model to preselected immune genes does not greatly diminish performance in predicting blood and psychosocial phenotypes, but the CTRA-based model does not appropriately reflect transcriptomic signatures of asthma and glucocorticoid phenotypes. This result suggests that genes outside the CTRA subset are important to many asthma and some psychosocial phenotypes.

## Replication of GxE effects

To validate the observed GxE effects on gene expression, we explored the overlap between the GxE genes and previously published datasets that measured interactions with different environments. We found that the majority of our GxE genes replicated in other datasets of GxE in gene expression ($p < 0.05$). Of our genes with GxE, 2 were found to have GxE in response to influenza (*Lee et al., 2014*), 59 were found to have GxE in response to a variety of chemical treatments (*Moyerbrailean et al., 2016*), 75 were found to have GxE in response to a variety of *Mycobacterium tuberculosis* (*Barreiro et al., 2012*), 87 were found to have GxE in response to rhinovirus (*Çalışkan et al., 2015*), 99 genes were found to have GxE in response to pathogens (*Nédélec et al., 2016*), and 117 had GxE with cell-type fraction in whole blood (*Kim-Hellmuth et al., 2020*).

