## [Decision Letter]

**Acceptance summary:**

This paper identifies transcriptional signatures of 19 variables including psychosocial factors, blood cell composition and asthma symptoms through RNA-sequencing and a new machine learning strategy. The results show that immune gene expression mediates the link to negative psychosocial experiences.

**Decision letter after peer review:**

Thank you for submitting your article "Psychosocial experiences modulate asthma-associated genes through gene-environment interactions" for consideration by *eLife*. Your article has been reviewed by 2 peer reviewers, and the evaluation has been overseen by a Reviewing Editor and Mone Zaidi as the Senior Editor. The reviewers have opted to remain anonymous.

The reviewers have discussed the reviews with one another and the Reviewing Editor has drafted this decision to help you prepare a revised submission.

Summary

This paper studies the interaction between genetics and psychosocial environment. RNA-seq in PBMCs is performed to assess associations between transcription, questionnaire response data, PBMC cytokine production, and pulmonary function tests. Additionally, the investigators utilized genetic data to assess gene by environment interactions. This work would be of interest to a wide viewership. However, significant issues with underlying methodology detract from interpretation of the data and make it difficult to draw any definitive conclusions. Given these underlying concerns, this paper's conclusions are not supported by the data presented and significant changes in methodology are necessary.

Essential revisions:

1. The greatest concern with this manuscript is that instead of using observed variables the use generalized linear models with elastic net regularization was employed to develop surrogate markers for observed variables within transcriptional data. This in itself is not inherently problematic if the authors were able to identify transcriptional signatures that explained a significant amount of variance in the observed variable. Unfortunately, the data suggests that transcriptional signatures only predict differences in peripheral leukocyte population. The % variance explained for neutrophils, lymphocytes, and eosinophils was >30% (not suggested as a cutoff). Notably, these three variables had significant and strong correlations at follow-up time points. This suggests that the authors indeed did identify transcriptional signatures that might reasonably serve as surrogates to these observed variables. However, the main focus of this manuscript is the psychosocial environmental exposure. The % variance explained for each these transcriptional signatures is < 5%. Further, these transcriptional signatures showed weak rho values when assessed at follow-up. These are not reasonable surrogates for observed values as these signatures could not within a reasonable margin of error predict many of these psychosocial measures. Due to this issue the interpretation of figures 2 and 3 becomes obscured. It remains usure what it means to have self-disclosure signature correlated with a FEV1 percent predicted signature when these signatures are not predictive of observed data. Do the observed values correlate? If so this might lend credence to authors proposed transcriptional signatures. Further the same issue plagues to the findings in figure 3 where these transcriptional signatures are used for the regression to test if known eQTLs have interaction with psychosocial measures. Since Figures 2 and 3 represent the major findings of the paper. Unfortunately, definitive conclusion cannot be drawn until the authors rectify this issue.

2. With regard to the mediation analysis presented in Figure 2, two issues arise. First, the authors present the mediation as if the three leukocyte populations were assessed simultaneously as mediators which does not appear to be the case from the description in the methods. Second, a sensitivity analysis should be performed with the mediation to assess if there is an observed confounder.

3. There is concern if the sequencing genomes at 0.1x coverage is sufficient. This leaves a significant amount data that is imputed. How many of the significant findings were derived from imputed values? Does the pair-wise error rate reported for genotype call between 0.03 and 0.12 refer to percent of base pairs where there is a discrepancy? Units are important, is this 3-12% or 0.03-0.12%.

4. It is unclear whether the interpretation of the eQTLs associated with asthma/atopy risk in population where 100% subjects have asthma. In these populations changes in gene expression can't correspond to risk because no individual is at risk; they already have the disease.

5. One lingering question is the clinical implication(s) of this extremely elegant work. In the big picture, how could we employ these results to tailor treatments for kids with asthma?

6. The effect of sex differences. While the authors recruited a similar number of male and females, it is not indicated their proportion in the 119. Additionally, asthma severity in females is exacerbated due to menstruation (Also, asthma is worsened during hormonal changes in females (e.g., menstruation, see Zein, J.G., Erzurum, S.C. Asthma is Different in Women. Curr Allergy Asthma Rep 15, 28 (2015). https://doi.org/10.1007/s11882-015-0528-y). According to the provided data, the authors do not correct for this effect and based on the current text, it is not address whether it is was taken into account. Additionally, what is the justification for dropping all the gene chromosomes? Would the results change if they were not removed?

7. One of the main difficulties while reading the text for me was that some of the topics are not clearly introduced and explained in the main text, while they were super easy to follow the same concepts and ideas in the supplementary information and in the appendix. Could the author modify the text to make it a little bit clearer, please?

8. The authors indicate that SES transcriptional signature "strong overlap with each other". Regarding this sentence: (a) SES is a very complex item to define and measure. Could the author specify how did they calculate it, e.g., income, education, a conglomerated index? (b) Is it a correlation between which variables, e.g., normalized RNA-seq counts, outputs from the model? Or you are talking about common present/absent transcripts? (c) Based on the heatmap, there are a total of 4 SES-houses unoccupied, subjective SES, parental income and houses rated >=fair, and only houses unoccupied and houses rated >= fair have a strong negative correlation, which one would have expected. Also, only parental income and houses rated >=fair are in the same cluster. (d) "For example, subjective SES was significantly correlated with objective parental responsiveness, family conflict, and self-reported self-disclosure, which is the extent to which the youths talk about their thoughts and feelings (r=-0.26, p=0.004, r=0.25, p=0.006, r=0.53, p=6.8*10-10, respectively)". Are those correlations between the transcriptional signatures associated with each variable? Or between the variables themselves. Could you clarify it, please?Sometimes the authors employed the term transcriptional signatures of subjective SES while other times they just referred to subjective SES. Finally, were those correlations models corrected by age? Did you look at associations with race/ethnicity? Did you correct from age, sex, race, ethnicity, SES?

9 "Using mediation analysis, we found significant (p<0.05) paths through all three blood composition signatures, such that, at the molecular level, self-disclosure association with higher pulmonary function could be partially explained by an increase in the proportions of monocytes and neutrophils and reduced proportions of lymphocytes (Figure 2b)". Were other physiological inputs (e.g., subjected SES), mediators (e.g., IL13 GC resistance) and/or outputs (e.g., asthma severity) modeled for the mediation analysis? If not, may the authors explain the reasoning of selecting the chosen variables, please?

10. "To examine the genotype-by-environmental effects… entire cohort of 251 individuals" I am concerned whether the subjects that were employed to identify the transcriptional signatures include representative of those individuals not included in the models in terms of socio-demographic characteristics. Also, the models were validated with only in a subset of psychological measurements-note that psychological measurements were indeed the worst predicted in the validation set. Therefore, I am not sure that the predicted psychological values are adequate. Is there any manner that we can verify their accuracy?

11. For the physiological analysis, why anxiety measurements were not incorporated, e.g., GAD-7 or similar, as they can trigger asthma symptoms?

---

## [Author Response]

SummaryThis paper studies the interaction between genetics and psychosocial environment. RNA-seq in PBMCs is performed to assess associations between transcription, questionnaire response data, PBMC cytokine production, and pulmonary function tests. Additionally, the investigators utilized genetic data to assess gene by environment interactions. This work would be of interest to a wide viewership. However, significant issues with underlying methodology detract from interpretation of the data and make it difficult to draw any definitive conclusions. Given these underlying concerns, this paper's conclusions are not supported by the data presented and significant changes in methodology are necessary.

We thank the reviewers for their insightful and excellent comments. We have carefully considered them and addressed all the major points. To address the comments, we have collected additional data on 31 variables for samples for which we did not have this information curated in our earlier submission (increasing the sample size on average by 80%). We also re-ran the eQTL mapping analysis because we realized that we had mis-annotated the regulatory region for some genes; yet this did not substantially change the results. We also removed one transcriptional signature (parental income), because upon further inspection the variable needed to be recoded and the transcriptional signature was not significantly predictive.

Essential revisions:1. The greatest concern with this manuscript is that instead of using observed variables the use generalized linear models with elastic net regularization was employed to develop surrogate markers for observed variables within transcriptional data. This in itself is not inherently problematic if the authors were able to identify transcriptional signatures that explained a significant amount of variance in the observed variable. Unfortunately, the data suggests that transcriptional signatures only predict differences in peripheral leukocyte population. The % variance explained for neutrophils, lymphocytes, and eosinophils was >30% (not suggested as a cutoff). Notably, these three variables had significant and strong correlations at follow-up time points. This suggests that the authors indeed did identify transcriptional signatures that might reasonably serve as surrogates to these observed variables. However, the main focus of this manuscript is the psychosocial environmental exposure. The % variance explained for each these transcriptional signatures is < 5%. Further, these transcriptional signatures showed weak rho values when assessed at follow-up. These are not reasonable surrogates for observed values as these signatures could not within a reasonable margin of error predict many of these psychosocial measures. Due to this issue the interpretation of figures 2 and 3 becomes obscured. It remains usure what it means to have self-disclosure signature correlated with a FEV1 percent predicted signature when these signatures are not predictive of observed data. Do the observed values correlate? If so this might lend credence to authors proposed transcriptional signatures. Further the same issue plagues to the findings in figure 3 where these transcriptional signatures are used for the regression to test if known eQTLs have interaction with psychosocial measures. Since Figures 2 and 3 represent the major findings of the paper. Unfortunately, definitive conclusion cannot be drawn until the authors rectify this issue.

We understand the reviewer's concern and thank them for the suggestions. Our objective is not simply to predict the unobserved values, but we also assume that the observed variable has high levels of noise and the measured value does not reflect the true biological effect. Therefore, we use the predicted values for all participants, including those for whom the variable was directly measured (denoising). The best fit for the model is used to predict, for both observed and unobserved samples, the biological impact on gene expression of the relevant variable. Please note that Supplementary File 1c contains cross-validated percent variance explained – the most stringent metric for evaluating model performance. Correlations between all variables and their transcriptional signatures were significant (pvalue<0.05), but we decided to use a cutoff of 1% cross-validated variance explained to only use the strongest transcriptional signatures. When we focus on these variables, the correlations between measured variables and their transcriptional signatures are significantly positive and range from 0.250.98 (mean=0.74).

We now also include correlations between measured and imputed values on a bigger subset of individuals (additional individuals not used in training the transcriptional signatures) in Supplementary File 1c. All correlations are significant (p<0.05) and range from 0.14-0.90 (mean=0.48).

The major limitation in predicting changes at follow up has been that for some variables the actual measured value did not change at all or very minimally in this short time frame.

Irrespective of being a cell-type composition variable, or a different type of variable if there was a large change and the signature was predictive, we were able to predict the change from the gene expression data.

To address the reviewer’s concern that correlations between transcriptional signatures may not reflect correlations between observed values, we have now added a Figure 1—figure supplement 2. This figure compares the correlation between the observed measurements and the correlation between the imputed/denoised variables. Some of the values for the pairwise correlations between transcriptional signatures slightly changed because we have now used the values imputed for the entire sample. Strong pairwise correlations in the measured variables are preserved in the imputed variables and detected by both approaches (Correlation of pair-wise correlation coefficients is 0.9, p-value = 6.069e-11). Additionally, pairwise correlations that are detected as significant in both the measured and the denoised/imputed variables are all in the same direction. Across all pairwise correlation coefficients, the correlation of those obtained between the measured variables or the imputed variables is 0.58, p-value < 2.2e-16. In our opinion, the fact that we detect a higher number of significant correlations in the imputed variables is due to the denoising effect.

We have added the following text to the manuscript:

Results:

"Overall, correlations between transcriptional signatures reflect correlations between measured variables (Figure 1—figure supplement 2), yet they are stronger between the transcriptional signatures, highlighting the denoising effect"

Discussion:

“Interestingly, we detected a higher number of significant correlations across imputed variables compared to the measured variables. This observation supports the effectiveness of the denoising procedure that we used to define transcriptional signatures.”

2. With regard to the mediation analysis presented in Figure 2, two issues arise. First, the authors present the mediation as if the three leukocyte populations were assessed simultaneously as mediators which does not appear to be the case from the description in the methods. Second, a sensitivity analysis should be performed with the mediation to assess if there is an observed confounder.

We agree with the reviewer that more complex scenarios may explain the observed correlations between the transcriptional signatures. Acknowledging these effects and given that the major focus of the manuscript is on the GxE effect in gene expression and asthma, we have decided to remove this section of the results as it would require additional analyses beyond the scope of this manuscript.

3. There is concern if the sequencing genomes at 0.1x coverage is sufficient. This leaves a significant amount data that is imputed. How many of the significant findings were derived from imputed values? Does the pair-wise error rate reported for genotype call between 0.03 and 0.12 refer to percent of base pairs where there is a discrepancy? Units are important, is this 3-12% or 0.03-0.12%.

We thank the reviewer for this comment. We used an external provider (Gencove, New York, NY) with a well-established genotyping-from-sequencing pipeline (PMID: 33743587, PMID: 33536225) to obtain genotype calls from high purity DNA samples for all participants. The actual coverage used was 0.4x, we have updated the methods to report this value. The genotype accuracy at this sequencing depth is 98.22% positive agreement and 99.82% negative agreement compared to genotyping arrays. In our study, we used the pairwise discordance between DNA and RNA-derived genotypes to identify sample swaps. The 3-12% DNA/RNA discordance was assessed only on the biallelic SNPs; yet only a subset of these SNPs is used for eQTL mapping (MAF>0.1 and within 1Mb of the TSS). Additionally, the reported figure represents an overestimate of the true error rate, as heterozygous genetic variants are inherently more prone to call errors and issues such as allele-specific expression may obscure genotype calls derived from RNA-seq data (known RNA editing sites were not excluded). Furthermore, we do not use discrete genotypes for eQTL mapping, and we use allelic dosage instead because they are less sensitive to genotype calling errors.

The following changes have been made in the methods section:

"Genotype data. All individuals in this study were genotyped from low-coverage (~0.4X) whole-genome sequencing and imputed to 37.5 M variants using the 1000 Genomes database by Gencove (New York, NY). […] This rate is not a direct measure of genotype call accuracy, but it is useful to identify possible sample swaps and mismatches."

4. It is unclear whether the interpretation of the eQTLs associated with asthma/atopy risk in population where 100% subjects have asthma. In these populations changes in gene expression can't correspond to risk because no individual is at risk; they already have the disease.

We agree with the reviewer that our cohort of case-only asthmatic participants would not allow us to determine which genes may predispose to higher asthma risk. That is why we used an external dataset from a Transcriptome-Wide Association Study (Zhang, 2019), which assessed disease-gene expression associations based on eQTL data from GTEx and multiple large GWAS for asthma risk in the typical case control study design. We hypothesized that the same genes and pathways that contribute to asthma risk are also involved in asthma symptom severity through gene regulatory variation in immune cells. We now clarify these concepts as reported below:

Results

"We hypothesized that genes and pathways that contribute to asthma risk are also involved in asthma symptom severity through gene regulatory variation in immune cells. We investigated whether genes associated with the risk for asthma were modulated through psychosocial experiences (E), and/or GxE effects."

Discussion

"As we gain additional knowledge on the mechanisms connecting psychosocial experiences to disease, these results can be useful to support the need for social interventions that may ultimately lead to improved overall health[…] The 124 instances of GxE with psychosocial experiences are particularly relevant when evaluating a polygenic risk score for asthma phenotypes."

5. One lingering question is the clinical implication(s) of this extremely elegant work. In the big picture, how could we employ these results to tailor treatments for kids with asthma?

We agree with the reviewer that this is indeed an extremely relevant question. Our work is at the intersection between clinical, molecular, and social studies. While genetic risk for extreme asthma phenotypes cannot be modified, our work shows that the social environment can modulate these important genetic effects on the regulation of genes relevant for asthma and immunity. We envision that this and other studies may open the road to social interventions aimed at improving the psychosocial environment of asthmatic children in order to ameliorate their symptoms and ultimately lead to better clinical outcomes. These social interventions may be implemented independently or together with drug treatments.

The following changes were made in the revised discussion:

"As we gain additional knowledge on the mechanisms connecting psychosocial experiences to disease, these results can be useful to support the need for social interventions that may ultimately lead to improved overall health. […] These social interventions may be implemented independently or together with drug treatments, and their impact could be further monitored through gene expression with larger longitudinal samples in future studies."

6. The effect of sex differences. While the authors recruited a similar number of male and females, it is not indicated their proportion in the 119. Additionally, asthma severity in females is exacerbated due to menstruation (Also, asthma is worsened during hormonal changes in females (e.g., menstruation, see Zein, J.G., Erzurum, S.C. Asthma is Different in Women. Curr Allergy Asthma Rep 15, 28 (2015). https://doi.org/10.1007/s11882-015-0528-y). According to the provided data, the authors do not correct for this effect and based on the current text, it is not addressed whether it is was taken into account. Additionally, what is the justification for dropping all the gene chromosomes? Would the results change if they were not removed?

We agree with the reviewer that sex is an important biological factor to consider. The proportion of males:females is similar in the subset of participants used for building the transcriptional signatures (68:51=1.33) as in the overall cohort (148:103=1.44). The male skew in our cohort reflects the higher predisposition of boys towards asthma compared to girls. We correct for sex effect, but not for menstrual cycle effects, because this variable was not collected in the ALOFT study. Additionally, our sample size would not allow us to correct for too many variables. To account for unmeasured confounders, which may also include the menstrual cycle, we correct for gene expression PCs in the eQTL mapping analysis.

Sex chromosome gene expression data need to be treated with caution as females may have higher expression of X chr gene escaping X inactivation. Y chr gene can of course only be assessed in males, reducing the overall sample size. For these reasons we decided to exclude sex chromosomes from our analyses. Of the 2695 asthma GWAS hits in the GWAS Catalog (ebi.ac.uk/gwas) none have been reported on the Y chromosome and only four associations (across two variants) are reported on the X chromosome.

7. One of the main difficulties while reading the text for me was that some of the topics are not clearly introduced and explained in the main text, while they were super easy to follow the same concepts and ideas in the supplementary information and in the appendix. Could the author modify the text to make it a little bit clearer, please?

We apologize for the lack of clarity. We have now moved the section "Transcriptional signatures aid in denoising the data." from the appendix to the Methods section as detailed below:

"Imputation and de-noising of transcriptional signatures. We assume that the observed variables have high levels of noise and the measured values do not reflect the true biological effects. […] Overall, our procedure should not create any biases but rather decrease the variability of the imputed/denoised variables thus reducing the chance of false positive GxE."

8. The authors indicate that SES transcriptional signature "strong overlap with each other". Regarding this sentence: (a) SES is a very complex item to define and measure. Could the author specify how did they calculate it, e.g., income, education, a conglomerated index? (b) Is it a correlation between which variables, e.g., normalized RNA-seq counts, outputs from the model? Or you are talking about common present/absent transcripts? (c) Based on the heatmap, there are a total of 4 SES-houses unoccupied, subjective SES, parental income and houses rated >=fair, and only houses unoccupied and houses rated >= fair have a strong negative correlation, which one would have expected. Also, only parental income and houses rated >=fair are in the same cluster. (d) "For example, subjective SES was significantly correlated with objective parental responsiveness, family conflict, and self-reported self-disclosure, which is the extent to which the youths talk about their thoughts and feelings (r=-0.26, p=0.004, r=0.25, p=0.006, r=0.53, p=6.8*10-10, respectively)". Are those correlations between the transcriptional signatures associated with each variable? Or between the variables themselves. Could you clarify it, please?Sometimes the authors employed the term transcriptional signatures of subjective SES while other times they just referred to subjective SES. Finally, were those correlations models corrected by age? Did you look at associations with race/ethnicity? Did you correct from age, sex, race, ethnicity, SES?

We agree with the reviewer that this section of the manuscript can be further clarified. We generally refer to correlations between transcriptional signatures, unless we specify the use of the measured variable. Importantly, the correlations between transcriptional signatures are also reflected in the measured variables, as described in our response to comment 1.

Subjective SES is self-reported by the parent using the MacArthur Socioeconomic Status Ladder (Adler et al., 2000) on a scale of 1-10 adjusted so that 1 corresponds to the lowest status and 10 to the highest. Details on this and all the other variables are reported in Supplementary file 2. We have now made it clear throughout the text whenever we refer to the individual measure of Subjective Socioeconomic status by always using the full variable name.

The following text is now included in the methods:

"The parent reported demographics, including their annual income and education level, and completed measures of subjective socioeconomic status (the McArthur ladder ((Adler et al. 2000), adjusted so that 1 corresponds to the lowest status and 10 to the highest), neighborhood stress ((Ewart and Suchday 2002), conflict with their child (the Parental Environment Questionnaire ((Elkins, McGue, and Iacono 1997)), and depressive symptoms (the CES-D ((Radloff 1977))."

The transcriptional signatures were calculated on normalized count data after regressing out the following confounding factors: RIN, percent reads mapping to exons, percent non-duplicate reads, site of RNA extraction, library preparation batch, sample collection wave, age, sex, height, weight, genotype PC1, genotype PC2, genotype PC3. These details are reported in the methods as below:

"First, we normalized the count data using the voom function in the *limma* v3.38.3 package in R(Law et al. 2014). Second, we regressed out the following confounding factors: RIN, percent reads mapping to exons, percent non-duplicate reads, site of RNA extraction, library preparation batch, sample collection wave, age, sex, height, weight, genotype PC1, genotype PC2, genotype PC3."

9. "Using mediation analysis, we found significant (p<0.05) paths through all three blood composition signatures, such that, at the molecular level, self-disclosure association with higher pulmonary function could be partially explained by an increase in the proportions of monocytes and neutrophils and reduced proportions of lymphocytes (Figure 2b)". Were other physiological inputs (e.g., subjected SES), mediators (e.g., IL13 GC resistance) and/or outputs (e.g., asthma severity) modeled for the mediation analysis? If not, may the authors explain the reasoning of selecting the chosen variables, please?

We agree with the reviewer that more complex scenarios may explain the observed correlations between the transcriptional signatures. Acknowledging these effects and given that the major focus of the manuscript is on the GxE effect in gene expression and asthma, we have decided to remove this section of the results as it would require additional analyses beyond the scope of this manuscript.

10. "To examine the genotype-by-environmental effects… entire cohort of 251 individuals" I am concerned whether the subjects that were employed to identify the transcriptional signatures include representative of those individuals not included in the models in terms of socio-demographic characteristics. Also, the models were validated with only in a subset of psychological measurements-note that psychological measurements were indeed the worst predicted in the validation set. Therefore, I am not sure that the predicted psychological values are adequate. Is there any manner that we can verify their accuracy?

We thank the reviewer for the suggestion. We have now included socio-demographic data separately for the sub-cohort used for training the transcriptional signature models (N=119), and for the entire cohort (N=251) in Supplementary File 1a. There are no differences in socio-demographic measures between the group used for training the transcriptional signature models and the group not included in the model for: sex ratio (chi^2=0.05, p-value=0.83), self-reported ethnicity (chi^2=2.43, p-value=0.49), parent’s income (Wilcoxon-Mann-Whitney W=2982.5, pvalue=0.98), or parent’s education (WilcoxonMann-Whitney W=3658, pvalue=0.07). These results are now included in Supplementary File 1a.

For this revised manuscript, we have collected additional data on 31 variables for samples for which we did not have this information curated in our earlier submission, and we show that GxE eQTLs detected using the transcriptional signatures are enriched for GxE eQTLs detected using the measured variables.

Note that with the measured variables we have limited power due to the smaller sample size and the inherent noise for some of these variables.

These new findings are reported in the Results section:

"To validate these GxE results we expanded the sample size for all variables with a transcriptional signature. We found that the GxE eQTLs detected with the measured variables (Figure 3-Source Data 7) were significantly enriched for low p-values in the GxE eQTLs detected with the transcriptional signatures (Figure 3—figure supplement 1, Figure 3A) and the interaction effects were highly significantly correlated (Supplementary File 1g)."

And in the methods:

"We calculated the enrichment of GxE genes from the measured variables (p-value<0.01) in the set of GxE genes from the transcriptional signatures (p-value<0.01). […] Additionally, we calculated the correlation of the standardized interaction effect size (z-score) for each gene obtained when considering measured variables and corresponding transcriptional signatures."

11. For the physiological analysis, why anxiety measurements were not incorporated, e.g., GAD-7 or similar, as they can trigger asthma symptoms?

We thank the reviewer for this comment. The ALOFT study did not collect GAD-7 or other measures of acute anxiety. Previous studies have shown that while both anxiety and depression are associated with worse asthma-related quality of life, depression is also associated with worse asthma control (Kim, 2006, PMID: 17035436). In our analysis we included child depressive symptoms (Children's Depression Inventory, Kovacs, 1992), but we did not find a significant transcriptional signature.